# Performance of Citric Acid-Bonded Oriented Board from Modified Fibrovascular Bundle of Salacca (*Salacca zalacca* (Gaertn.) Voss) Frond

**DOI:** 10.3390/polym13234090

**Published:** 2021-11-24

**Authors:** Luthfi Hakim, Ragil Widyorini, Widyanto Dwi Nugroho, Tibertius Agus Prayitno

**Affiliations:** 1Department of Forest Product Technology, Faculty of Forestry, Universitas Sumatera Utara, Jl. Tri Dharma Ujung No. 1, Medan 20155, Indonesia; 2JATI-Sumatran Forestry Study Analysis Center, Universitas Sumatera Utara, Jl Tri Dharma Ujung No. 1, Medan 20155, Indonesia; 3Department of Forest Product Technology, Faculty of Forestry, Universitas Gadjah Mada, Jl. Agro No. 1, Yogyakarta 55281, Indonesia; rwidyorini@ugm.ac.id (R.W.); wdnugroho@ugm.ac.id (W.D.N.); ta_prayitno@ugm.ac.id (T.A.P.)

**Keywords:** fibrovascular bundle, salacca frond, modified orientation board, citric acid

## Abstract

The fibrovascular bundle (FVB) in palm plants consists of fiber and vascular tissue. Geometrically, it is a long fiber that can be used as an oriented board raw material. This research aimed to examine the performance of citric acid-bonded orientation boards from modified FVB salacca frond under NaOH + Na_2_SO_3_ treatment and the bonding mechanism between the modified FVB frond and citric acid. The results showed that changes in the chemical composition of FVB have a positive effect on the contact angle and increase the cellulose crystallinity index. Furthermore, the mechanical properties of the oriented board showed that 1% NaOH + 0.2% Na_2_SO_3_ with 60 min immersion has a higher value compared to other treatments. The best dimension stability was on a board with the modified FVB of 1% NaOH + 0.2% Na_2_SO_3_ with 30 and 60 min immersion. The bonding mechanism evaluated by FTIR spectra also showed that there is a reaction between the hydroxyl group in the modified FVB and the carboxyl group in citric acid. This showed that the modified combination treatment of NaOH+Na_2_SO_3_ succeeded in increasing the mechanical properties and dimensional stability of the orientation board from the FVB salacca frond.

## 1. Introduction

The fibrovascular bundle (FVB) is a tissue in monocot plants, specifically palm plants, that consists of xylem, phloem, sclerenchyma fibers, and parenchyma tissue. According to Hakim et al. [1], the FVB has good enough mechanical properties to be used as a raw material for making composite boards. Its advantages as a new natural fiber include being biodegradable, renewable, low in cost, environmentally friendly, light in weight, and abundant in nature. However, the FVB also has disadvantages such as high water absorption and anisotropic properties, low durability, and compatibility with some conventional matrices as well as adhesives [2,3].

Several types of research studies have been carried out on natural fibers using a chemical treatment to modify the surface structure and improve the compatibility between adhesive and raw material. This includes the use of alkali modification in natural fibers due to the convenience and simplicity of its application [4,5]. Previous research studies have used sodium hydroxide (NaOH); however, the combination of NaOH+Na_2_SO_3_ as a chemical for modification has not been widely reported. Therefore, the use of NaOH+Na_2_SO_3_ as a chemical modification exposes more –OH groups on the fiber surface and increases compatibility with adhesives since the surface structure facilitates higher interface bonding as raw material for composite boards [6,7]. Alkali and sodium sulfite modification with low concentration aims to expand cellulose and dissolve amorphous components in hemicellulose and lignin so that it can expose more –OH groups, increase the accessibility of crystalline cellulose, increase hydrophilic properties, and reduce sugar content so as to increase the adhesion bond between cellulose and adhesive. The bond between the modified FVB and the adhesive improves with the presence of more –OH groups due to the combined modification of NaOH+Na_2_SO_3_.

The use of modified FVB as a raw material for composite boards is more suitable when it is developed into an orientation board product [8]. Geometrically, it is a long fiber; therefore, compactness in orienting the raw materials is one of the advantages for improving mechanical properties and dimensional stability. Oriented board or oriented strand board (OSB) is defined as a three-layer structural board produced by strand material bonded with a thermosetting type of exterior adhesive and additives to increase dimensional stability such as water-resistant resin [9]. A previous study on the mechanical and physical properties of cement-bonded OSB by Papadopoulos et al. [10] showed that the geometry of raw materials affects OSB quality. Hassani et al. [11] improved the mechanical and physical properties of oriented strand lumber (OSL) by using fortification of nano-wollastonite on urea formaldehyde resin and stated that this treatment could improve dimensional stability and activate the bonds with cellulose hydroxyl groups. However, in this study, orientation boards made from modified FVB were manufactured using citric acid as a natural adhesive. This study focuses on the use of citric acid as a binder to FVB modified by alkali and sodium citrate in the manufacture of orientation boards.

Natural adhesives such as citric acid are used as an alternative in the manufacture of environmentally friendly composite boards [12,13,14,15,16,17,18,19]. However, research on the manufacture of oriented boards from modified FVB frond combined with citric acid as adhesive and the bonding mechanism between the modified FVB by NaOH+Na_2_SO_3_ combination and the citric acid adhesive has not been reported. Therefore, this research aims to evaluate the performance of citric acid-bonded oriented boards from modified FVB *S. zalacca* under NaOH+Na_2_SO_3_ treatment.

## 2. Materials and Methods

### 2.1. Materials

The fibrovascular bundle extracted from the *S. zalacca* frond was used as a raw material for the manufacture of oriented board. The extraction was based on Hakim et al.’s study [20], and the bundles were cut 25 cm in length before chemical treatment (Figure 1). The chemical solution was modified using sodium hydroxide (NaOH) anhydride and sodium sulfite (Na_2_SO_3_). Meanwhile, the citric acid (anhydrous) used as adhesive without further purification was supplied by Rudang Chemical Company (Indonesia).

### 2.2. Chemical Treatments and Chemical Content Changes

Before the board was manufactured, the fibrovascular bundle was chemically modified with NaOH and Na_2_SO_3_ and their combination, as shown in Table 1. The chemical characteristics analyzed after treatment were α-cellulose and hemicellulose determined based on ASTM D1103-60, Klason lignin, as well as ash content determined based on ASTM D110-84 and ASTM D1102-84, respectively.

### 2.3. Contact Angle and Mechanical Properties of Modified Fibrovascular Bundles

The contact angle was measured using a method by Schellbach et al. [21]; meanwhile, two parallel fibrovascular bundles separated by 1–2 mm were attached to the sample holder and observed under the microscope. Furthermore, water was dripped between the two fibrovascular bundles to obtain a liquid that hangs, while the image of the liquid was photographed with a stereo optical camera and analyzed using IC-Measure version 2.0.0.245 to measure the contact angle.

Meanwhile, the mechanical properties of the modified fibrovascular bundle were evaluated based on the ASTM D-3379-75 standard. The FVB with 8–12% moisture content was cut to a length of about 90 mm + 0.1 mm and fixed on a 30 mm long paper frame using epoxy adhesive (ALF Epoxy Adhesive, P.T. Alfaglos, Semarang, Indonesia). The measurements were conducted in 50 replicates for each chemical treatment. The mechanical properties were determined using a universal testing machine (UTM Tensilon RTF 1350, Tokyo, Japan), with a 1 mm/min crosshead speed. Before the test, the supporting paper’s middle part was cut out, and the test mounting was carried out according to the method proposed by Hakim et al. [1,20] (Figure 2). In addition, the density of the modified FVB was measured based on the method described by Munawar et al. [22].

### 2.4. Index Crystallinity of Modified FVB

The crystallinity index (CrI) was determined by considering the regions of crystalline and amorphous cellulose. Crystalline cellulose is determined at a 2θ peak in the reflection plane position (*I*_002_) (maximum intensity between 22.5° and 23°), while amorphous cellulose (*I*_am_) is the minimum-intensity position (between 18° and 19°). Furthermore, the diffraction spectra were obtained at room temperature (20–22 °C) from radiation generated by Maximax X-ray Diffractometer-7000 (Shimadzu, Japan). The measurements were carried out at 40 kV and 20 mA with a detector placed on the range of 2θ from 5° to 70° at a scan speed of 2°/min. Subsequently, the percentage crystallinity index (%CI) of the cellulose was calculated using the formula provided by Segal et al. [23].
$$\%\mathrm{CI}=\frac{\left(I_{002}-I_{am}\right)}{I_{002}}\times 100$$
where *I*_002_ is the maximal peak intensity at a 2θ angle of approximately 22°–23° and *I*_am_ is the minimum peak intensity (amorphous region) at a 2θ angle of about 18°–19°.

### 2.5. The Manufacture of Oriented Board

After modification treatment, FVB was air-dried (10% moisture content) before being used as a raw material for orientation boards. The board was to be manufactured with dimensions of 250 × 250 × 8 mm^3^ with a target density of 0.8 g/cm^3^. The citric acid solution used as the adhesive was dissolved in water to a concentration of 60%. The resin content of citric acid is 30%. The adhesive solution was sprayed onto the raw materials and oven-dried at 75 °C for more than 8 h to reduce the moisture content to 4–6%. The fibrovascular bundle was manually hand-formed into a three-layer oriented mat using a forming box. Three different orientations (0°, 45°, and 90°) of the board were prepared for each fibrovascular bundle, each in three layers, namely face:core:back = 30%:40%:30% weight ratio (Figure 3). The three-layer oriented mat was hot-pressed at 180 °C for 10 min and a pressure of 3 MPa at three replications.

### 2.6. Evaluation of Oriented Board Properties

All of the samples were conditioned at environmental conditions for approximately a week to reach a moisture content of ±6% before the measurement of the oriented board properties according to the Japanese Industrial Standard for particleboard (JIS A 5908). The mechanical properties of the oriented board were measured for modulus of rupture (MOR), modulus of elasticity (MOE), the strength of the internal bond (IB), and screw holding power (SHP) using a universal testing machine (Tensilon RTF 1350, Tokyo, Japan), while the physical properties were tested for thickness swelling (TS) and water absorption (WA). The static bending samples (MOR and MOE) of 200 × 50 × 8 mm^3^ were treated by the three-point bending method under dry conditions with a span distance of 150 mm and a crosshead speed of 10 mm/min. The IB specimens of 50 × 50 × 8 mm^3^ were randomly measured from both surfaces of each specimen. Furthermore, the SHP specimen of 75 × 50 × 8 mm^3^ was drawn two positions from the board of each sample with a speed of 2 mm/min, while the TS and WA of 50 × 50 × 8 mm^3^ were evaluated by water immersion for 24 h at 20 °C.

### 2.7. Fourier Transform Infrared (FTIR) Spectroscopy

The fibrovascular bundle-oriented board samples were previously soaked in boiling water for 2 h to remove unreacted citric acid, conditioned in room temperature water for 1 h, dried at 35 ± 5 °C for 12 h, and powdered through a 100-mesh screen. FTIR spectroscopy was performed at room temperature (approximately 25 °C) using the FTIR-4200 spectrophotometer (8201PC-Shimadzu, Tokyo, Japan) and the KBr disk method with 12 cm^−1^ of resolution to determine the assignment of absorbance bands to specific functional groups.

## 3. Results and Discussion

### 3.1. Properties Change of Modified FVB

Table 2 shows the contact angles, densities, crystallinity index, and chemical composition of modified FVB. The modified FVBs’ α-cellulose content increased from 43.11% (treatment A0-30) to 53.38% (treatment C12-30) and slightly decreased again from treatments C12-60, C14-30, and C14-60 (50.71%, 39.62%, and 35.15%, respectively).

The increase in cellulose from A0-30 to C12-30 treatment does not imply an increase in cellulose content, but rather that the other components, hemicellulose and lignin, decreased. Meanwhile, the α-cellulose fraction reduces during treatments C12-60, C14-30, and C14-60 because the α-cellulose component dissolves. This reduction was caused by the NaOH + Na_2_SO_3_ concentration, which has the capacity to destroy amorphous cellulose at high concentrations and longtime immersion. Similar to the previous research, 3% NaOH treatment + steaming on *S. zalacca* FVB frond yielded 54.53% cellulose [24], and a 15% NaOH treatment on areca palm frond fiber (*Dypsis lutescens*) raised the cellulose content to 63.45% [25].

The hemicellulose content of modified FVB decreased from 32.24% with the 30 min water immersion treatment to 21.91% with the treatment modified with 1%NaOH + 0.4% Na_2_SO_3_ and 60 min immersion. In contrast to previous studies, the alkaline treatment of 3% on the FVB of the frond of *S. zalacca* became 74.09% [24]; 5% NaOH treatment on banyan tree root fibers decreases hemicellulose to 10.74% [26]. Hemicellulose is one of the components of lignocellulosic materials that is easily degraded under alkaline treatment because the structure of the hemicellulose components is nonlinear and more amorphous than crystalline [27,28].

The lignin component of the modified *S. zalacca* frond FVB decreased from 28.1% (treatment A0-30) to 18.7% (treatment C14-60). This pattern of decline is relatively the same as that in previous studies using modified alkali [24,25,26,29,30]. Furthermore, the NaOH + Na_2_SO_3_ treatment was more effective in maintaining α-cellulose, hemicellulose, and lignin content compared to previous studies by Darmanto et al. [31] using *S. zalacca* as raw material with higher concentration and high-energy steam.

The modified FVB had the highest density value of 0.40 g/cm^3^ under the water treatment with 30 min immersion. Meanwhile, the lowest was 0.35 g/cm^3^ under 1% NaOH + 0.4% Na_2_SO_3_ with 60 min immersion. This decreased density value is due to the degradation factor of the chemical components during the modification treatment, which causes the weight of FVB decrease.

The contact angle was measured to determine when the modified FVB is easily wetted by a liquid; meanwhile, the value obtained is 24.02° with treatment H (Table 2), which is much lower than the contact angle of bamboo, jute, ramie, and kenaf [4,32]. The modification of the combination of NaOH+Na_2_SO_3_ decreased the contact angle compared to the water immersion treatment. Furthermore, it was affected by the surface roughness due to changes in the structure of the chemical composition of the surface [33]. Surface changes due to the degradation of chemical composition and accessibility of hydroxyl groups on the surface of FVB are the two factors affecting the level of wettability. The NaOH+Na_2_SO_3_ treatment led to the removal of low molecular weight material such as hemicellulose and lignin, while the cellulose content increased. Based on previous research, the alkali modification reduced some impurities and wax in the FVB surface [5]. The results were almost similar to those of 3% NaOH with steaming treatment on the FVB of *S. zalacca* [24], 5.5% NaOH treatment of fiber-empty fruit bunches of oil palm [34], 15% NaOH treatment of palm fiber areca [25], 5% NaOH treatment of *Tridax procumbens* fiber [29], 5% NaOH treatment of oil palm mesocarp fiber [35], and 5% NaOH treatment of banyan tree root fiber [26].

Table 3 shows the mechanical properties of modified FVB. The higher maximum load of modified FVB was 122.57 ± 9.94 N after A0-60 treatment, while the lower maximum load was 57.15 ± 6.86 N after C14-60 treatment. Interestingly, the tensile strength increased from A0-30 to C12-60 treatment but decreased after C14-30 and C12-60 treatment. Conversely, the increase in tensile strength was not accompanied by the increase in maximum load. This phenomenon is explainable because modified FVB’s tensile strength increases due to reduction in the FVB’s transversal area. Furthermore, this reduction was due to the surface degradation during the modification treatment.

Darmanto et al. [24] reported that a combination of 1% NaOH and high temperature with 2 bars pressure treatment resulted in a 355 MPa increase in the tensile strength of FVB *S. zalacca* frond. Compared to the result of Darmanto’s research, this study was more effective and efficient because high temperatures and pressures were not used. Young’s modulus has the same pattern as the tensile strength of the modified FVB. The higher Young’s modulus of modified FVB increased from water treatment to C12-30 treatment but slightly decreased after C12-60, C14-30, and C14-60 treatment, respectively.

The mechanical properties of modified FVB are influenced by the material’s crystallinity index. Figure 4 shows the X-ray diffraction spectra of modified FVB. The crystallinity index (CrI) of modified FVB increased from A0-60 treatment to C12-60 treatment before decreasing again after C14-60 treatment. The crystallinity index of modified FVB immersed in water for 60 min was 53.04%, whereas for the 1% NaOH treatment counterpart, it increased to 56.25%. In addition, the crystallinity index of FVBs subjected to 1% NaOH + 0.2% Na_2_SO_3_ treatment with 60 min immersion increased to 58.33%, whereas for the counterparts treated with 1% NaOH + 0.4% Na_2_SO_3_ with 60 min immersion, it decreased to 47.43%. The mechanical properties of fiber are affected by the crystallinity index. The higher the crystallinity index of a fiber, the higher is the predicted fiber strength [36]. The treatment of 1% NaOH + 0.2% Na_2_SO_3_ with 60 min immersion resulted in an optimal increase in the crystallinity index of modified FVB. This occurs due to the degradation of amorphous components, which include hemicellulose, lignin, and wax. These chemical components are sensitive to alkali, so they are easily removed by alkaline modification treatment [37]. The 1% NaOH + 0.4% Na_2_SO_3_ treatment with 60 min immersion decreased the crystallinity index because cellulose polymer is penetrated by sulfite ions (SO_3_^−^), so some of the cellulose and lignin components begin to dissolve [38].

### 3.2. Modulus of Rupture (MOR) and Modulus of Elasticity (MOE) of a Modified Orientation Board

The MOR and MOE values of the oriented board were observed in two positions, namely perpendicular (ꓕ) and parallel (//). Generally, their values for an oriented board with 0° orientation angle were higher than those of the board with orientation angle 45° and 90°. The MOR value in the perpendicular position was higher than the MOR value in the parallel position for all orientation angles (0°, 45°, and 90°), which is similar to the previous research conducted by Munawar et al. and Baharin et al. [39,40]. Therefore, the MOR value increased with the combined treatment with 1% NaOH + 0.2% Na_2_SO_3_ and decreased with the combined treatment with 1% NaOH + 0.4% Na_2_SO_3_.

As shown in Figure 5, the highest MOR and MOE values of the orientation board are 23.4 MPa (ꓕ) and 11.4 MPa (//) with orientation angle 0°, 15.5 MPa (ꓕ) and 13.5 MPa (//) with orientation angle 45°, and 17.7 MPa (ꓕ) and 16.9 MPa (//) with orientation angle 90°, respectively. All of these highest values were achieved in the combination treatment of 1% NaOH + 0.2% Na_2_SO_3_ with 60 min immersion, whereas the lowest values were achieved in the 30 min water immersion treatment, and they were 11.8 MPa (ꓕ) and 4.9 MPa (//) with orientation angle 0°, 7.9 MPa (ꓕ) and 7.3 MPa (//) with orientation angle 45°, and 7.6 MPa (ꓕ) and 7.5 MPa (//) with orientation angle 90°, respectively.

The highest MOE values were obtained at 7.4 GPa (ꓕ) and 4.0 GPa (//) for 0° orientation angle, 4.7 GPa (ꓕ) and 4.1 GPa (//) for 45°, and 5.2 GPa (ꓕ) and 5.1 GPa (//) for 90°, respectively, in the combination treatment of 1% NaOH + 0.2% Na_2_SO_3_ for 60 min. Meanwhile, the lowest MOE values were obtained with the water immersion treatment for 30 min at 3.3 GPa (ꓕ) and 1.2 GPa (//) for 0°, 2.1 GPa (ꓕ) and 2.0 GPa (//) for 45°, and 2.5 GPa (ꓕ) and 2.2 GPa (//) for 90°, respectively.

Umemura et al. [41] reported that wood waste particleboard with citric acid as an adhesive has an MOR value of 10.7 MPa. Another study found that pressing at a temperature of 180 °C during the manufacture of particleboard using citric acid as an adhesive increases the MOR value by 19.6 MPa [42]. Furthermore, Liao et al. [17] stated that the addition of sucrose to the citric acid adhesive during the manufacture of low-density particleboard gives an MOR value of more than 6.0 MPa. The manufacture of orientation boards from natural fibers such as *Sansevieria* using phenol–formaldehyde (PF) adhesive also increases the MOR value by 403 MPa [39]. In this research, a board made using 30% citric acid adhesive and a compression temperature of 180 °C successfully produced an oriented board with an MOR value of 23.4 MPa. This occurred because the raw material in the form of long fibers subjected to alkali modification treatment can increase the mechanical value of the composite board obtained [43]. This shows that raw materials in the form of long natural fibers for the use of composite boards will result in good mechanical properties for the boards [44,45]. Meanwhile, alkali modification of long fibers such as FVB can increase the accessibility of cellulose, hemicellulose, and lignin to increase the FVB interface bond and citric acid adhesive, which will affect the strength of the composite board [46,47].

The research by Santoso et al. [48] on the manufacture of particleboard using particles of *Nypa fruticans* frond that bonded citric acid and maltodextrin adhesives give an MOE of 0.5 GPa. Meanwhile, Widyorini et al. [42] found that the addition of maltodextrin and the compression step to the particleboard made from the particles of salacca frond gives an MOE of 4.1 GPa. The research conducted by Kusumah et al. [18] using sorghum particles (bagasse) as a raw material for particleboard with citric acid adhesive gave an MOE of 5.27 MPa. Meanwhile, this research gave the highest MOE value of 8.2 MPa, which is higher than the previous results. The dimensions of the raw material in the form of long fibers influenced the mechanical properties of the composite board obtained. The perpendicular position test with the regular orientation of the FVB arrangement can increase compactness when subjected to a load.

### 3.3. Strength of Internal Bond (IB)

The IB increased from water treatment for 30 min to the combination treatment of 1% NaOH + 0.2% Na_2_SO_3_ and then decreased again at the 1% NaOH + 0.4% Na_2_SO_3_ treatment for 30 min. The orientation board with an angle of 0° has a greater IB value than boards with angles of 45° and 90°. The IB value of the orientation board made from the modified FVB frond of *S. zalacca* is shown in Figure 6.

The highest IB value of the board was found in the combination treatment of 1% NaOH + 0.2% Na_2_SO_3_ with 30 min immersion and 1% NaOH + 0.2% Na_2_SO_3_ for 60 min immersion, and it was 0.46 MPa (0°) and 0.35 MPa (90°), respectively, while the lowest was 0.29 MPa (0°) and 0.25 MPa (90°), respectively, with 1% NaOH + 0.4% Na_2_SO_3_ treatment with 60 min immersion.

Moreover, the IB values of 90° and 45° orientation boards are lower than 0° because the 90° orientation boards are arranged crosswise, consisting of three layers of the face, core, and back for the bonding area and contact between the fibers on the surface of one layer with another layer to reduce. Some of the values in this research were still higher—0.37 MPa [49] and 0.15 MPa [50,51]—than the previous ones that used waste wood particles and citric acid as adhesives.

According to Munawar et al. [39], the orientation board of *Sansevieria* fiber bonded using phenol–formaldehyde (PF) had an IB value of 1.33 MPa, which is much higher compared to that of this study. Walther et al. [52] also stated that the IB value of the orientation board made from kenaf glued with PF is 0.44 MPa (with orientation angle 0°), which is lower compared to that of this study.

### 3.4. Screw Holding Power (SHP)

The value of SHP increased from control to the combination treatment of 1% NaOH + 0.2% Na_2_SO_3_ with 30 min immersion; however, it decreased again on the combined treatment of 1% NaOH + 0.4% Na_2_SO_3_ with 30 min immersion. The results showed that the oriented board with 0° and 90° angles of orientation have relatively similar SHP values. The highest SHP values were 519 N (0°), achieved in the combined treatment of 1% NaOH + 0.2% Na_2_SO_3_ with 60 min immersion, and 504 N (45°) and 495 N (90°), achieved in the combined treatment of 1% NaOH + 0.2% Na_2_SO_3_ with 30 min immersion, respectively. The lowest values of HPS were 318 N (0°), 307 N (45°), and 293.5 N (90°), achieved in the combined treatment of 1% NaOH + 0.4% Na_2_SO_3_ with 60 min immersion. The histogram of the SHP of the oriented board is shown in Figure 7.

This research showed that the modified 1% NaOH + 0.2% Na_2_SO_3_ treatment with 60 min immersion was the best in influencing the screw holding strength. According to Hung et al. [53], the adhesion between the modification raw material and the adhesive increases with the SHP. Meanwhile, another factor that affects the SHP is the size and dimensions of the raw material. This is because the longer and thicker the raw material, the more is the SHP value of the composite board [54]. Compared to the use of particle raw material, the SHP value of the oriented board is higher than the particleboard with the same adhesive (citric acid) [19]. Finer particles are less effective in transferring the stress from particle to particle compared to FVB. Similarly, damage to the board due to screw threads occurs faster in particle raw materials compared to long-fiber raw materials.

The single-layer orientation board has a higher SHP value than the three-layer orientation board, while the 0° orientation angle is slightly higher than 45° and 90°. This showed that the orientation angle of 0° is a single-layer board, while 45° and 90° are a three-layer board.

### 3.5. Thickness Swelling (TS) and Water Absorption (WA)

The orientation board has the highest WA value in the water immersion treatment with 30 min immersion, which is 51.18% (0°), 54.87% (45°), and 55.26% (90°). By contrast, the lowest WA value is 43.50% (0°) in the 1% NaOH treatment with 30 min immersion, and 16.20% (45°) and 47.80% (90°) in the combination treatment of 1% NaOH + 0.2% Na_2_SO_3_ for 60 min. In addition, the orientation board has the highest TS value of 19.64% (0°), 21.65% (45°), and 22.65% (90°) in the 30 min water immersion treatment, while the lowest value is 14.99% (0°) and 16.20% (45°) in the 1% NaOH + 0.2% Na_2_SO_3_ with 60 min immersion, and 17% (90°) in the 1% NaOH + 0.2% Na_2_SO_3_ with 30 min immersion. Generally, the WA and TS values decreased from treatment A (30 min water immersion) to treatment F (1% NaOH + 0.2% Na_2_SO_3_ with 60 min immersion) and increased again in treatment G (1% NaOH + 0.4% Na_2_SO_3_ with 30 min immersion). The WA and TS values of the modified orientation board are shown in Figure 8.

The results have lower WA and TS values compared to the previous research conducted by Kemalasari and Widyorini [55], which stated that the WA and TS of particleboard with the raw material of salacca frond particles with citric acid adhesive ranged from 36–62.26% to 6.95–22.42%, respectively. Widyorini et al. [42] also researched the manufacture of particleboard using salacca frond particles and citric acid as adhesives and found a WA of 54.2%. It was also reported that the extractive substances in the salacca frond particles do not affect the water absorption of the particleboard bound to the citric acid [19]. Meanwhile, Kusumah et al. [56], who used citric acid as an adhesive, reported that the use of 30% citric acid is effective in reducing the water absorption of composite boards. Based on the alkali modification in this research, the combined treatment of 1% NaOH + 0.2% Na_2_SO_3_ with 60 min immersion succeeded in reducing water absorption to 43.0%.

### 3.6. Fourier Transform Infrared (FTIR) Spectroscopy

The effect of modified NaOH and the combination of NaOH + Na_2_SO_3_ on the chemical structure of the FVB orientation board of the salacca frond bonded with citric acid is shown in Figure 9. Based on the results, the ester functional group was detected at the peak of the wavelength around 1733 cm^−1^, which shows the C=O stretching group, indicating the formation of carbonyl group or C=O group [16,19,42]. This showed a reaction/bond between the carboxyl group in citric acid and the hydroxyl group in the modified FVB. The peak with the highest intensity was discovered in treatment C12-30 (1% NaOH + 0.2% Na_2_SO_3_ for 30 min) and C12-60 (1% NaOH + 0.2% Na_2_SO_3_ for 60 min), while the modified treatment with a higher concentration of Na_2_SO_3_ (treatment C14-30 and C14-60) showed a decrease in intensity. This showed the mechanical properties of the orientation board, which has the best mechanical value in treatment C12-30 and C12-60.

Figure 10 shows the FTIR spectra of the modified FVB (FVB+C12-30 and FVB+C12-60) and the modified orientation board (board+C12-30 and board+C12-60). The modified FVB (FVB+C12-30 and FVB+C12-60) in the FTIR spectra did not show a peak at a wavelength of 1733 cm^−1^, but there was a significant peak on the modified orientation board (board+C12-30 and board+C12-60) at a wavelength of 1733 cm^−1^, which indicates the formation of a carbonyl group (C=O stretching) and an ester group C=O). This shows that there is a reaction between the hydroxyl group in the modified FVB and the carboxyl group in citric acid [42]. Furthermore, the formation of an ester bond on the modified orientation board made from modified FVB can increase the adhesive bond with the modified raw material. In addition, Menezzi et al. [57] described that esterification occurs between citric acid’s carboxylic acid activities and the many aromatic and aliphatic hydroxyl groups of cellulose and lignin.

As expected, orientation boards were manufactured from FVB modified with citric acid adhesive and their mechanical properties were tested. It was found that the combination of NaOH+Na_2_SO_3_ treatment influences the modulus of rupture, modulus of elasticity, internal bond, as well as screw holding power of the orientation board. The 1% NaOH + 0.2% Na_2_SO_3_ treatment with 30 and 60 min immersion increased the mechanical properties. As a structural panel, orientation boards made of modified FVB bonded with citric acid as a natural adhesive still rank below the wood-based composite with conventional adhesives. These orientation boards can be recommended as nonstructural panels such as partitions and insulation boards. In terms of the use of structural panels, FVB can be improved as a potential new raw material for the development of composite boards by combining it with other raw materials such as wood and veneer.

## 4. Conclusions

The research was conducted using the alkali modification of the fibrovascular bundle extracted from an *S. zalacca* frond for raw material of the orientation board. The results showed that the combined treatment of 1% NaOH + 0.2% Na_2_SO_3_ with 30 and 60 min immersion has a positive effect on increasing the mechanical properties of the orientation board. The 0° orientation angle also had a better effect on the MOR and MOE tests for the position perpendicular to the fiber. Furthermore, the bonding mechanism of FVB modified by the combination of 1% NaOH + 0.2% Na_2_SO_3_ by immersion for 30 and 60 min bonded citric acid adhesive showed good bonding quality. The FTIR spectra also showed the peak intensity of the wavelength at 1733 cm^−1^ with the detection of C–O stretching the carbonyl and ester groups, which indicated that there is a reaction between the hydroxyl group in the modified FVB and the carboxyl group in citric acid. Based on these results, the combination of 1% NaOH + 0.2% Na_2_SO_3_ treatment for 30 and 60 min immersion is successful in reducing water absorption and thickness swelling of the orientation board.

## Figures and Tables

**Figure 1 polymers-13-04090-f001:**
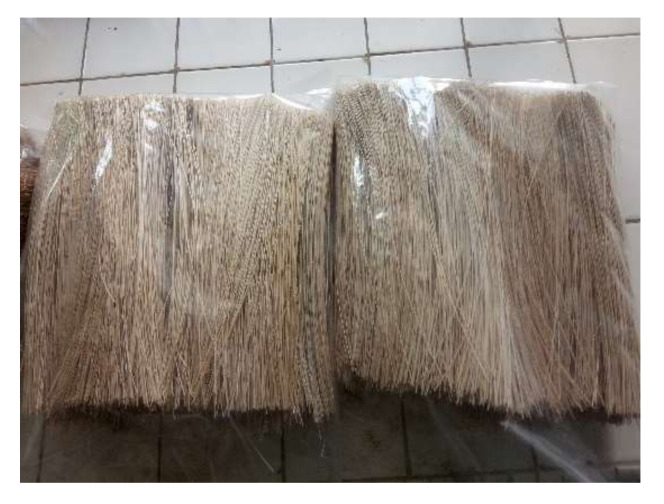
The fibrovascular bundle extracted from *S. zalacca*.

**Figure 2 polymers-13-04090-f002:**
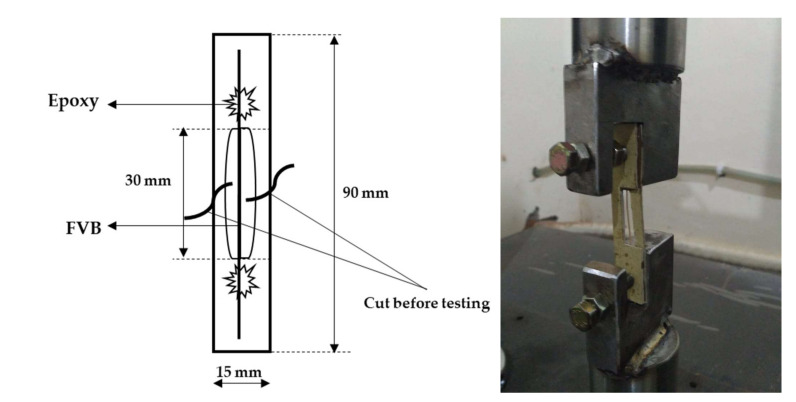
An illustration of FVB mechanical properties testing [20].

**Figure 3 polymers-13-04090-f003:**
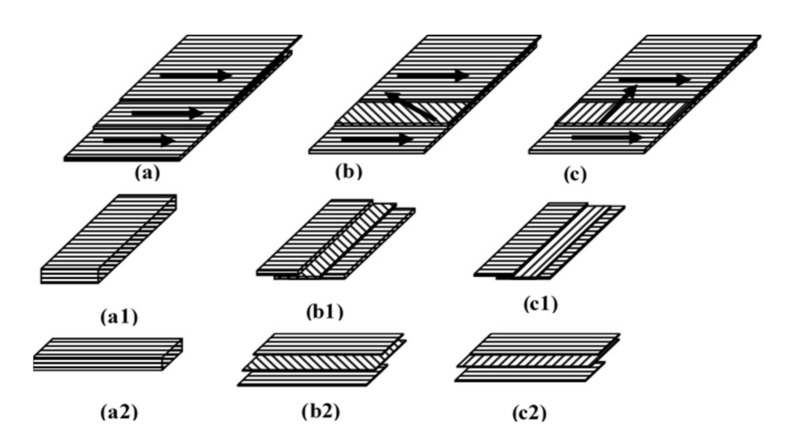
Layout for manufacturing fibrovascular bundle-oriented board. (**a**–**c**) are 0°, 90°, and 45° oriented, respectively. (**a1**–**c1**) are parallel (//) samples for the modulus of rupture (MOR) and modulus of elasticity (MOE). (**a2**–**c2**) are perpendicular (ꓕ) samples for MOR and MOE.

**Figure 4 polymers-13-04090-f004:**
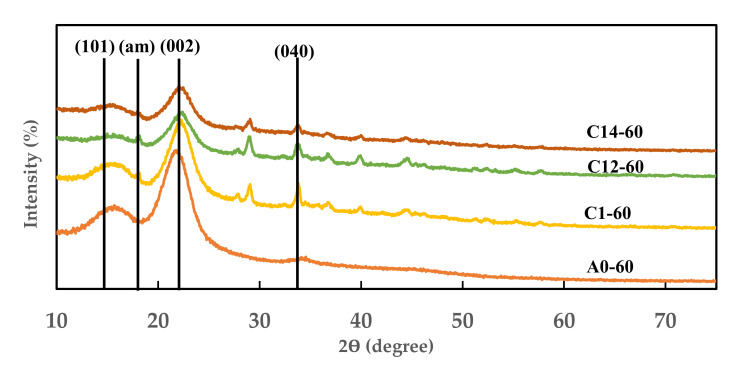
Diffractogram of X-ray diffraction of modified FVB. A0-60: water (60 min); C1-60: 1% NaOH (60 min); C12-60: 1% NaOH + 0.2% Na_2_SO_3_ (60 min); and C14-60: 1% NaOH + 0.4% Na_2_SO_3_ (60 min).

**Figure 5 polymers-13-04090-f005:**
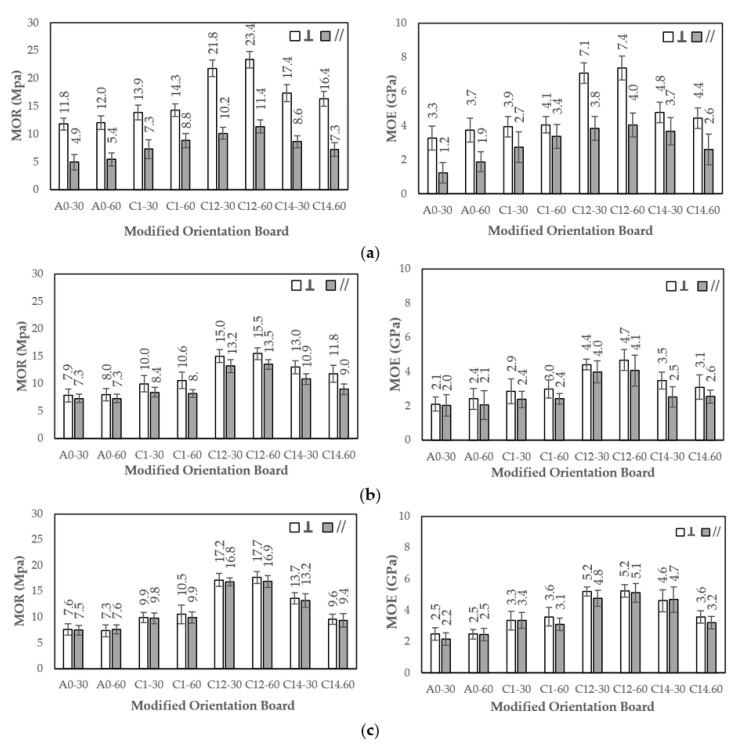
Histogram of MOR and MOE. (**a**–**c**) are 0°, 45°, 90°, respectively; ꓕ: perpendicular, //: parallel; A0-30: water (30 min); A0-60: water (60 min); C1-30: 1% NaOH (30 min); C1-60: 1% NaOH (60 min); C12-30: 1% NaOH + 0.2% Na_2_SO_3_ (30 min); C12-60: 1% NaOH + 0.2% Na_2_SO_3_ (60 min); C14-30: 1% NaOH + 0.4% Na_2_SO_3_ (30 min); and C14-60: 1% NaOH + 0.4% Na_2_SO_3_ (60 min). Error bar represents the standard deviation.

**Figure 6 polymers-13-04090-f006:**
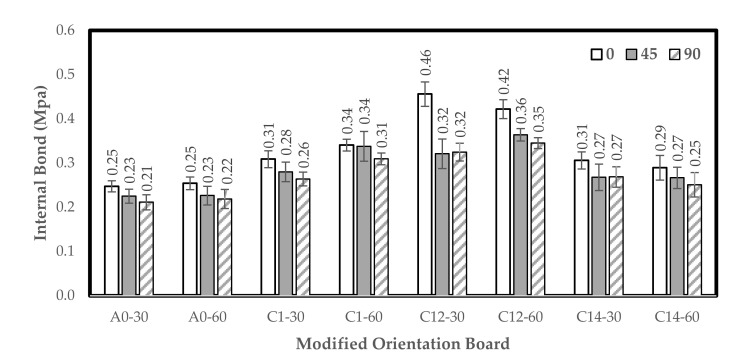
Histogram of the internal bond. A0-30: water (30 min); A0-60: water (60 min); C1-30: 1% NaOH (30 min); C1-60: 1% NaOH (60 min); C12-30: 1% NaOH + 0.2% Na_2_SO_3_ (30 min); C12-60: 1% NaOH + 0.2% Na_2_SO_3_ (60 min); C14-30: 1% NaOH + 0.4% Na_2_SO_3_ (30 min); and C14-60: 1% NaOH + 0.4% Na_2_SO_3_ (60 min). Error bar represents the standard deviation.

**Figure 7 polymers-13-04090-f007:**
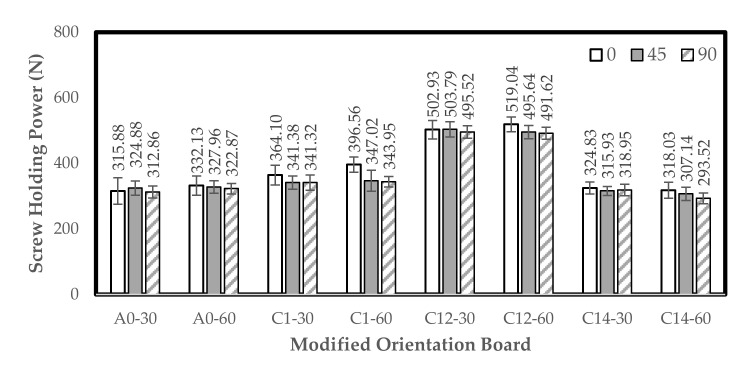
Histogram of screw holding power. A0-30: water (30 min); A0-60: water (60 min); C1-30: 1% NaOH (30 min); C1-60: 1% NaOH (60 min); C12-30: 1% NaOH + 0.2% Na_2_SO_3_ (30 min); C12-60: 1% NaOH + 0.2% Na_2_SO_3_ (60 min); C14-30: 1% NaOH + 0.4% Na_2_SO_3_ (30 min); and C14-60: 1% NaOH + 0.4% Na_2_SO_3_ (60 min). Error bar represents the standard deviation.

**Figure 8 polymers-13-04090-f008:**
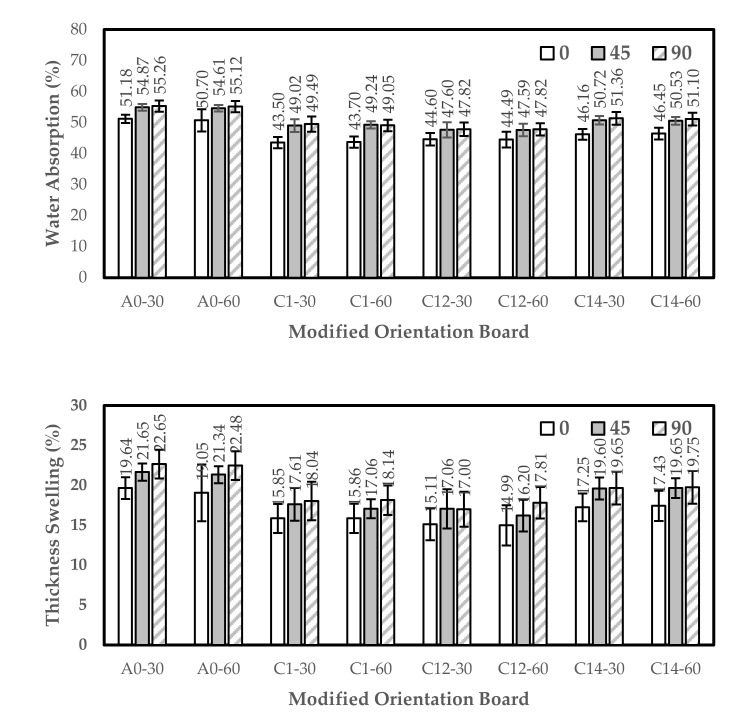
Histogram of water absorption and thickness swelling. A0-30: water (30 min); A0-60: water (60 min); C1-30: 1% NaOH (30 min); C1-60: 1% NaOH (60 min); C12-30: 1% NaOH + 0.2% Na_2_SO_3_ (30 min); C12-60: 1% NaOH + 0.2% Na_2_SO_3_ (60 min); C14-30: 1% NaOH + 0.4% Na_2_SO_3_ (30 min); and C14-60: 1% NaOH + 0.4% Na_2_SO_3_ (60 min). Error bar represents the standard deviation).

**Figure 9 polymers-13-04090-f009:**
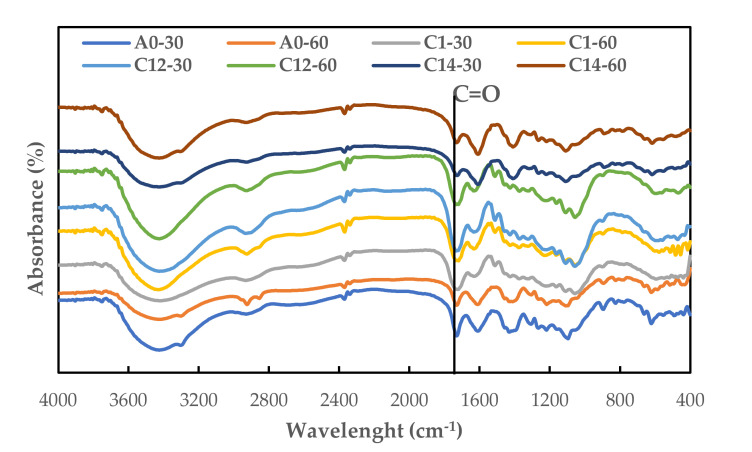
FTIR spectra of oriented board. A0-30: water (30 min); A0-60: water (60 min); C1-30: 1% NaOH (30 min); C1-60: 1% NaOH (60 min); C12-30: 1% NaOH + 0.2% Na_2_SO_3_ (30 min); C12-60: 1% NaOH + 0.2% Na_2_SO_3_ (60 min); C14-30: 1% NaOH + 0.4% Na_2_SO_3_ (30 min; and C14-60: 1% NaOH + 0.4% Na_2_SO_3_ (60 min).

**Figure 10 polymers-13-04090-f010:**
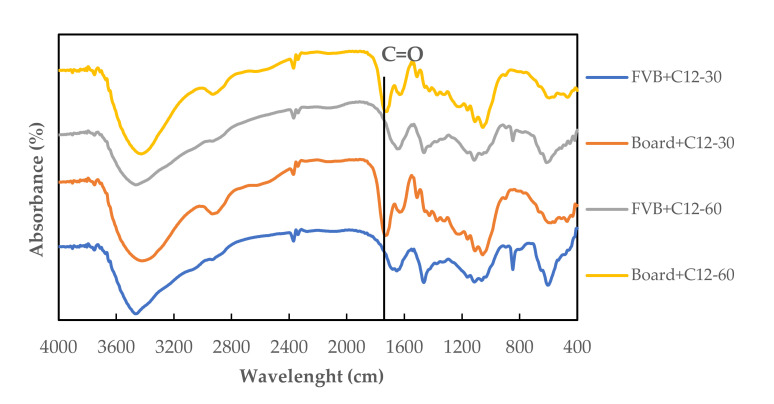
FTIR spectra of modified FVB (FVB+C12-30 and FVB+C12-60) and modified orientation board (board+C12-30 and board+C12-60). FVB+C12-30: modified fibrovascular bundle with 1% NaOH + 0.2% Na_2_SO_3_ (30 min); FVB+C12-60: modified fibrovascular bundle with 1% NaOH + 0.2% Na_2_SO_3_ (60 min); board+C12-30: oriented board with modified raw material (1% NaOH + 0.2% Na_2_SO_3_ (30 min); board+C12-60: oriented board with modified raw material (1% NaOH + 0.2% Na_2_SO_3_ (60 min).

**Table 1 polymers-13-04090-t001:** The chemical treatment of the fibrovascular bundle.

Treatment	NaOH+Na_2_SO_3_ Combination	Immersion Time (Minutes)
A0-30	Untreated, water immersion	30
A0-60	Untreated, water immersion	60
C1-30	1% NaOH	30
C1-60	1% NaOH	60
C12-30	1% NaOH + 0.2% Na_2_SO_3_	30
C12-60	1% NaOH + 0.2% Na_2_SO_3_	60
C14-30	1% NaOH + 0.4% Na_2_SO_3_	30
C14-60	1% NaOH + 0.4% Na_2_SO_3_	60

**Table 2 polymers-13-04090-t002:** Contact angle, density, crystallinity index, and chemical composition of modified FVB.

Treatment	Contact Angle (°)	Density (g/cm^3^)	CrI (%)	α-Cellulose (% wt)	Hemicellulose (% wt)	Lignin (% wt)
A0-30	92.30 ± 7.66	0.40 ± 0.09	–	43.11 ± 0.71	32.24 ± 0.87	28.18 ± 1.13
A0-60	84.47 ± 3.25	0.40 ± 0.08	53.04	43.13 ± 0.90	32.17 ± 0.79	28.09 ± 0.96
C1-30	55.83 ± 3.41	0.39 ± 0.12	–	46.45 ± 0.52	30.73 ± 0.94	27.11 ± 1.03
C1-60	46.20 ± 5.04	0.39 ± 0.17	56.25	48.15 ± 0.77	31.00 ± 1.57	26.33 ± 2.31
C12-30	44.38 ± 7.65	0.38 ± 0.06	–	53.38 ± 1.67	27.92 ± 0.76	22.05 ± 0.89
C12-60	34.88 ± 3.88	0.37 ± 0.18	58.33	50.71 ± 1.46	24.72 ± 0.66	21.33 ± 0.85
C14-30	31.98 ± 4.60	0.37 ± 0.09	–	39.62 ± 1.55	20.85 ± 0.69	21.22 ± 0.92
C14-60	24.02 ± 2.80	0.35 ± 0.12	47.43	35.15 ± 0.72	21.91 ± 0.98	18.78 ± 1.21

**Table 3 polymers-13-04090-t003:** Mechanical properties of modified FVB.

Treatment	Maximum Load (N)	Tensile Strength (MPa)	Young’s Modulus (GPa)
A0-30	121.01 ± 13.53	217.19 ± 28.11	2.23 ± 0.44
A0-60	117.28 ± 8.82	222.10 ± 18.63	2.23 ± 0.42
C1-30	117.68 ± 8.77	253.32 ± 25.81	2.52 ± 0.40
C1-60	122.57 ± 9.94	275.41 ± 21.28	2.29 ± 0.41
C12-30	119.92 ± 9.33	313.93 ± 41.43	2.55 ± 0.24
C12-60	110.30 ± 6.67	331.66 ± 33.38	2.28 ± 0.36
C14-30	77.83 ± 6.06	267.72 ± 39.52	2.18 ± 0.43
C14-60	57.15 ± 6.86	237.99 ± 43.45	1.91 ± 0.38

## Data Availability

The data presented in this study are available on request from the corresponding author.

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
