# Peer review of "Performance of Citric Acid-Bonded Oriented Board from Modified Fibrovascular Bundle of Salacca (Salacca zalacca (Gaertn.) Voss) Frond"

_polymers, 2021, doi:10.3390/polym13234090_

Round 1

Reviewer 1 Report

  • General: Chemical formulations should be consistent throughout the text. The number of atoms in all formulations is to be written in subscript format. Therefore, all cases of Na2SO3 are to be modified to Na2SO3. Unfortunately, even in the Abstract, both ways are mistakenly written.
  • Introduction: Oriented Boards (OB) and Oriented-Strand Boards (OSB) are getting wider attention in the past few decades or so, because various materials can be used to produce these boards. While the esteemed authors here have used fibrovascular bundles (FVB), other researchers have used a variety of raw materials, like poplar strands. I strongly recommend adding a short paragraph in the Introduction section, briefly discussing the latest achievements and studies on OSB, using nano- and micro-scaled wollastonite to improve different properties. In this connection, studies conducted by Prof. Antonios Papadopoulos, Prof. Ayoub Esmailpour, Prof. Jeff Morrell, Prof. George Mantanis, Engr. Vahid Hassani, and Dr. Roya Majidi are highly recommended. Some of them were even published in Polymers Journal.
  • Figure 4: As a researcher who has spent more than two decades on natural materials, I found it very hard to accept that some of the columns in this Figure (No. 4) have actually no error bars. Is it really so? If positive, I find it outstanding. For instance, Figure 4,a, MOR column D â”´, or Figure 4,a, MOE column F //, have no error bars. I do recommend the authors to once again check their results from the very scratch to make sure if the original numbers of the replications were correctly noted down.
  • Figure 9 (a, b, and c): The authors have used linear graph. However, and from a statistical point of view, I believe histograms (column graphs) are to be used here. Therefore, the esteemed authors are recommended to modify these three graphs, similar to what they have already used in Figures 4, 5, and 6.

I believe that other parts are written and discussed very well.

Author Response

Dear Rewiewer 1

 We Really appreciate your efforts in handling our manuscript, entitled “Performance of Citric Acid-Bonded Oriented Board from Modified Fibrovascular Bundle of Salacca (Salacca zalacca (Gaertn.) Voss) Frond”, with very constructive comments by the reviewers. The following attached are authors’ responses to the reviewers’ comments: (All modifications in the text have been made in yellow highlight)

Reviewer 2 Report

In my opinion the focus of this study is not related with Journal Polymers scopes. The author reported the effect of fiber orientation and the effect of fiber surface treatment. The mechanism of adhesion was not elaborated in details. The state of the art and background study is lacking. Overall, the English proviciency and readable of manuscript is low.

The title does not fit with the contain, the experimental design is weak, and there are some redundant and repetition pattern of discussion

In abstract, please put in the parentheses, the shorter of fibrovascular bundle (FVB)

The purpose of alkaline combined with sodium sulfite treatment in this study was not clear, both chemicals swell the fibers and dilute non cellulosic compounds of the fiber.

Line 136 page 4 : the immersion was conducted at 20 oC. How to adjust the temperature and why it should be at that temperature?

Figure 2, please improve the quality of the figure to make it readable

On the discussion, this study is only confirming their results with others author's result which in some cases the methodology is not comparable. Author should report their finding based on a critical analysis

Probably author must separate between the results that mostly only reading the datas and the discussion section

Line 260-270, do they reported the same method as author did?

The authors have been reported the previous study about the anatomy of Salacca, but in this case author’s extract the FVB. It is better to display picture of what is the FVB look likes

It is better to name the sample properly and briefly, for example A0-30, A0-60; C1-30, C1-60, C12-30, C14-30 etc

Figure 9, caption is not clear

Author Response

Dear Reviewer 2:

 We really appreciate your efforts in handling our manuscript, entitled “Performance of Citric Acid-Bonded Oriented Board from Modified Fibrovascular Bundle of Salacca (Salacca zalacca (Gaertn.) Voss) Frond”, with very constructive comments by the reviewers. The following attached are authors’ responses to the reviewers’ comments: (All modifications in the text have been made in yellow highlight)

Reviewer 3 Report

Dear editor, dear authors,

The article presented by Hakim et al. is a very interesting piece of research where the oriented strand boards from Salacca are glued with citric acid after chemical modification to produce bio-based panels which might be suitable in an optic of circular economy. The authors present a huge amount of experiment in a concise way and this render the content effective.

For this reason, I am generally favorable for the publication of this work, despite a few aspects need to be modified in order to render the paper even more understandable.

  • Please give further information about the strands itself after treatment, before gluing with citric acid. Are they dried, are they still wet?
  • Please give details on the sample which underwent mechanical properties (line 87)
  • I understood the authors apply 30% of resin on the board, but please separate in another sentence otherwise it might be confused with the concentration of the resin.
  • Be systematic, line 112 should be: 0, 45 and 90°
  • Table 2 has no standard deviation reported. Was it only one test? If yes, I understand, otherwise please provide the standard deviation.
  • Concept: when a lignocellulosic resource is undergone to alkaline attack, also the amorphous celluloses can be degraded and contribute to the increase of overall crystallinity degree
  • Figure 2 is of bad quality and does not bring further information to the contact angle of table 2. I suggest to removing it.
  • Thickness swelling paragraph should be considered before the chemical characterization which is the very last part to be presented.
  • Finally in the final part of the discussion, it would be nice have some comparison with other bio-based panels or even commercial products in order to better positionate this innovative product in the market.
  • Other smaller comments: Please report the acronym the first time it is find in the text FVB is already in the abstract. Line 70-71 correct with …was chemically modified… Line 396 oriented board (instead of orientation board).

Author Response

Dear Reviewer 3

 We really appreciate your efforts in handling our manuscript, entitled “Performance of Citric Acid-Bonded Oriented Board from Modified Fibrovascular Bundle of Salacca (Salacca zalacca (Gaertn.) Voss) Frond”, with very constructive comments by the reviewers. The following attached are authors’ responses to the reviewers’ comments: (All modifications in the text have been made in yellow highlight)

Round 2

Reviewer 2 Report

Line 234 page 6, C14-60, please use "dash" not "dot"

I still believe that this article should be published in Materials instead of Polymers